# Irrigation Effect on Yield, Skin Blemishes, Phellem Formation, and Total Phenolics of Red Potatoes

**DOI:** 10.3390/plants11243523

**Published:** 2022-12-14

**Authors:** Manlin Jiang, Tracy Shinners-Carnelley, Darin Gibson, Debbie Jones, Jyoti Joshi, Gefu Wang-Pruski

**Affiliations:** 1Faculty of Agriculture, Dalhousie University, Truro, NS B2N 5E3, Canada; 2Research, Quality & Sustainability, Peak of the Market, Winnipeg, MB R3H 0R5, Canada; 3Gaia Consulting Ltd., Newton, MB R0H 0X0, Canada

**Keywords:** anthocyanin, russeting, silver patch, *Solanum tuberosum*, suberized cell layer, surface cracking

## Abstract

Dark Red Norland is an important potato cultivar in the fresh market due to its attractive bright, red colour, and good yield. However, skin blemishes such as silver patch, surface cracking, and russeting can negatively influence the tuber skin quality and marketability. It is well known that potato is a drought-sensitive plant. This study was conducted to determine whether irrigation would affect Dark Red Norland’s yield and skin quality. A three-year field trial was conducted by Peak of the Market in Manitoba, Canada. Plants were treated under both irrigation and rainfed conditions. The results show that irrigation increased the total yield by 20.6% and reduced the severity of surface cracking by 48.5%. Microscopy imaging analysis demonstrated that tubers from the rainfed trials formed higher numbers of suberized cell layers than those of the irrigated potatoes, with a difference of 0.360 to 0.652 layers in normal skins. Surface cracking and silver patch skins had more suberized cell layers than the normal skins, with ranges of 7.805 to 8.333 and 7.740 to 8.496, respectively. A significantly higher amount of total polyphenols was found in the irrigated samples with a mean of 77.30 mg gallic acid equivalents (GAE)/100 g fresh weight (fw) than that of the rainfed samples (69.80 mg GAE/100 g fw). The outcome of this study provides a better understanding of the water regime effect causing these skin blemishes, which could potentially be used to establish strategies to improve tuber skin quality and minimize market losses.

## 1. Introduction

Potato (*Solanum tuberosum* L.) is the fourth most important food crop in terms of volume and consumption after maize, wheat, and rice in the world [1]. It is also one of the most studied crops in much of the latest literature [1,2,3]. About one-fifth of the potatoes grown in Canada are for the fresh table market [4]. Canada ranks as the fifth biggest fresh potato exporter in the world [4]. During 2020 and 2021, Canada exported CAD 319 million’s worth of fresh potatoes [4]. Norland cultivar was one of the top three registered seed potato varieties grown in Canada in 2020 [4]. Ιt is a common fresh market cultivar and popular because of its bright red colour. Dark Red Norland is a developed strain of the Norland cultivar, which has darker red skin colour and high yield but the same weaknesses: skin discoloration and skin blemish defects. Skin blemishes can badly affect tuber appearance and marketability.

Potato tubers are covered with a protective corky skin tissue called periderm. Periderm has a complex structure that is made up of three types of cells. The visible outermost layer of the skin is called the phellem layer, which is composed of many layers of suberized cells. Under the phellem is the phellogen, which is made of layers of meristematic cells. Under the phellogen is the phelloderm, which is made of layers of parenchyma-like cells [5]. Phellogen cells divide outwards to make suberized phellem; phellogen cells divide inwards to make phelloderm [6].

Potato periderm contains enzymes and metabolites that can respond to biotic and abiotic stresses. Phenolic compounds are the most abundant secondary metabolites in plants. Potato skin has higher amounts of phenolic compounds than potato flesh [7]. These compounds are important in plant defence mechanisms as antioxidants. The synthesis of phenolic compounds is induced in response to biotic and abiotic stimulation, such as drought, chilling, pathogens, or nutrient deficiency [8]. As a group of phenolic compounds, anthocyanin synthesis and accumulation in potato tissues are also considered indicators of stress resistance [9,10]. Anthocyanin discoloration in *Solanaceae* is more likely due to a change in the balance between anthocyanin biosynthesis and degradation [11]. The ability to enhance skin-set development and suberization can greatly reduce surface blemishes, shrinkage and flaccidity, blemishes, and infections [7].

‘Skin blemishes’ are those defects on tuber skin that can badly influence the tuber’s appearance and grading. Surface cracking, silver patch, and russeting are the three major skin blemishes found on Dark Red Norland tubers in the field. Surface cracking is seen as shallow, corky cracks on the tuber skin, normally presenting as rough, latticed areas of tuber skin (Figure 1a). Most cracks are generated when the internal pressure exceeds the tensile strength of the surface tissues during tuber enlargement, and the outer periderm bursts [12]. Silver patch is a defect that appears as silvery, smooth patches on the tuber skin (Figure 1b). This defect has not yet been described in the literature but has been given the name silver patch by Dr. Tracy Shinners-Carnelley (personal communication). Russeting presents as protruding dark-brown patches on the surface of tubers and is considered a defect when it occurs on tubers of smooth-skinned cultivars (Figure 1c). It negatively affects the protective functions of the skin, including the prevention of water loss and resistance to pathogen invasion [13]. It is believed that these defects are not associated with any disease-causing pathogens, since no pathogens have been isolated and identified in these defective tissues (Dr. Tracy Shinners-Carnelley, personal communication).

As a drought-sensitive plant, adequate soil moisture is suggested to be maintained at all stages of potato development [14]. Water stress may inhibit or even completely stop one or more physiological processes, such as transpiration, photosynthesis, cell enlargement, and enzymatic activities [15]. Limited irrigation at different stages of potato growth results in earlier crop maturity and decreases plant growth, tuber yield, the number of tubers per plant, and tuber size and quality [16,17]. Drought during the periods of tuber initiation and bulking has the most drastic effect on the yield [16]. Smaller tuber sizes and increased external defects were found in a previous study when the irrigation gradually declined and no irrigation occurred during the tuber initiation [18]. The effects of drought stress on tuber physiological development could include decreases in tuber number, increases in misshapen tubers, reduced tuber dry matter, and reduced water content [19].

Skin blemishes, such as surface cracking, silver patch, and russeting, significantly showed up on Dark Red Norland tubers for unknown reasons in 2019–2021 field trials in Manitoba. It is proposed that these blemishes were caused by environmental factors. This study determined the relationships between the water regime and the yield of Dark Red Norland and their tuber skin blemishes. In addition, phellem structure, total phenolics, and anthocyanin were studied to obtain a better understanding of these skin blemishes.

## 2. Results

### 2.1. Total Yield

The three-year data show that the highest yield of Dark Red Norland was in the medium size with a range of 2.25–3.0″ in diameter for approximately 71.2% of the total yield (Figure 2). The proportions of less than 2″, 2–2.25″, and 3–3.5″ of the total yield were 4.8%, 8.2%, and 14.7%, respectively. When the total yields (tubers of all sizes) were compared, higher yields were found in the irrigated plots, with a mean of 429.032 cwt/ac for the three years. Irrigation improved the proportion of the yield from medium- and large-size tubers (by 8.1% and 125.6%) and increased the proportion of the total yield by 20.6% but did not significantly change the yields of less than 2″ and between 2 and 2.25″. 

### 2.2. Skin Blemishes

Silver patch appeared with the highest percentage of occurrence (nearly 50%) in both the rainfed and irrigated plots (Figure 3). A significantly higher percentage of surface cracking was found in the rainfed plots (27.39%) compared to the irrigated (18.45%), which increased by 48.5%. A lower percentage of russeting was found in the rainfed plots (10.06%) compared to the irrigated plots. (17.98%) (Figure 3). Normal tubers accounted for only 22.72% and 24.79% of the rainfed and irrigated treatments, respectively.

### 2.3. Suberized Cell Layer

The representative samples of suberized skins in the normal skin type and the three defected skin types of surface cracking, silver patch, and russeting are shown in Figure 4. Normal skin had organized cells, which were well-packed (Figure 4a); russeting skin had rough skin surface and irregular cells (Figure 4d); silver patch and surface cracking had obviously more suberized cell layers (Figure 4b,c). Silver patch skin had well-arranged cells (Figure 4c), while surface cracking had cracks between the cells (Figure 4b).

In regard to normal skins, the tubers of the rainfed plots had more layers of suberized cells compared to those of the irrigated plots in all three years (Table 1). The differences in the suberized cell layers in the normal skins between the two water regimes were in a range of 0.360–0.652 during the three years. Among the four skin types, surface cracking and silver patch skins had significantly more suberized cell layers than those of normal and russeting skins. This situation occurred in most of the plots in all three years (Table 2). Table 2 lists eight treatments with heat stress applied at different growth stages, including ‘Tuber Initiation’, ‘Tuber Bulking’, ‘Tuber Skinset’, and ‘No Stress’ in both the rainfed and irrigated plots. We do not discuss the heat treatment in this paper because the heat treatment did not have significant effects on the soil temperature in the field trial. However, we could clearly distinguish the differences in the number of suberized cell layers among the four skin types. Surface cracking and silver patch skins had the most suberized cell layers with ranges of 7.805 to 8.333 and 7.740 to 8.496, respectively.

### 2.4. Total Phenolic Content

Tubers from 32 plots in the 2020 and 2021 field seasons were analyzed for the total phenolic content. In both years, silver patch skin and russeting skin showed more total phenolics than normal skin (Figure 5). The amounts of total phenolic content in the silver patch and russeting skin samples were 76.85 mg GAE/100 g fw and 77.63 mg GAE/100 g fw, respectively. The normal skin had the lowest amount of total phenolics, with an average of 69.12 mg GAE/100 g fw. In addition, the irrigated samples had more total phenolics (77.30 mg GAE/100 g fw) than the rainfed samples (69.80 mg GAE/100 g fw).

### 2.5. Anthocyanin Content

Thinly sliced ‘normal’ and ‘silver patch’ skin samples were observed under the bright field of a microscope (Figure 6). In Figure 6A–J, normal skin samples are shown in the left column, which clearly show more pinkish pigments in the periderm. Silver patch skin samples are shown in Figure 6K–T in the right column, showing a less red colour; instead, there is a layer of a brown-coloured compound in the skin cells. The comparisons between the normal and silver patch skin samples demonstrated that the loss of the reddish pigment in the silver patch skin was the reason for the blemish. Therefore, the anthocyanin contents were measured in the normal and silver patch skin tissues. The anthocyanin content was significantly lower in the silver patch skin tissues compared to the tissues of the normal skin type. Figure 7 shows a summarized analysis, including all the data from both the 2020 and 2021 seasons. Within two years, the normal skins had a higher total anthocyanin content, with an average of 0.0624 mg C3GE/100 mg fresh weight than that of the silver patch skins, which had an average of 0.0444 mg C3GE/100 mg fresh weight. These results demonstrate that the silver patch skins lost a significant amount of anthocyanin.

## 3. Materials and Methods

Three-year (2019–2021) field trials were conducted by Peak of the Market (POM) in Manitoba, Canada. All the laboratory experiments of this study were conducted at the Faculty of Agricultural, Dalhousie University located in Truro, Nova Scotia, Canada.

### 3.1. Three-Year Field Trial

The field trial to produce the potato cultivar, Dark Red Norland (*Solanum tuberosum* L.), for this study was conducted at the POM Research Site in Winkler, Manitoba, by Gaia Consulting (https://gaiaconsulting.mb.ca/) (accessed on 10 December 2022). There were 32 plots each year. Half of the plots were treated without irrigation, called ‘rainfed’ plots, while the others were treated with irrigation. The irrigation was applied using a lateral irrigation system (Figure 8). The hand-feel method was used to determine if the water holding capacity was close to or below 70%, which meant it was time to irrigate the plots. The irrigation schedule was different in each of the three years due to the local daily precipitation, soil moisture evaporation, and the amount of water storage for irrigation. The dates and the amount of applied irrigation each year are shown in Table 3. In 2019, irrigation was applied 11 times in the field trial, while there it was only applied 6 and 4 times in 2020 and 2021, respectively. It should be mentioned that the water reservoir ran dry on 9 July 2021 and no additional irrigation water could be applied after that in the 2021 field trial.

The tuber yield and tuber blemish defects, including surface cracking, silver patch, and russeting, were recorded. After harvest, tubers from the differently treated plots were rated for skin colour, external blemishes, yield, and size. The harvested tubers were separated into 5 groups based on size, which were <2″, 2–2.25″, 2.25–3.0″, 3–3.5″, and >3.5″. Tubers with different skin blemishes were counted and transformed into percentage numbers in each size group by using the formula:Number of defected tuber/Total number of graded tubers × 100(1)

The grading rules were based on the Peak of the Market Pre-Pack Inspection Manual (POM, 2010). Selected tubers were shipped to Dr. Wang-Pruski’s lab at Dal AC for all the lab analyses.

### 3.2. Sampling

#### 3.2.1. Sampling for Suberin Analysis

The tubers were washed and graded after harvesting each year by Gaia Consulting. Four medium-sized tuber samples were randomly selected from each plot and sent to Dal AC for suberin analysis each year. Thirty-two bags of tubers were received each year and stored in a cooler at 4 °C and 90% relative humidity (RH). Each bag was checked and typical skin types, including normal, surface cracking, silver patch, and russeting, were marked on the tubers (Figure 1). Tubers were photographed under bright light before cutting. Skin samples were taken based on the four types of skin blemish occurrence in each bag and processed using the methods described later.

#### 3.2.2. Sampling for Total Phenolics and Anthocyanin Analyses

Eight tubers were selected from each plot for total phenolics and anthocyanin analyses in both the 2020 and 2021 seasons. Half of the samples (four tubers) had bright red colour and relatively normal skins, while the other half (four tubers) were selected with one or more skin blemishes of surface cracking, silver patch, and/or russeting. These tubers normally had lighter skin colour (less red) (Figure 9a).

Skin samples were collected from the eight tubers based on normal, surface cracking, silver patch, and russeting skin types for each bag of samples. Each type of skin sample was cut off from at least two tubers in a bag by a knife with a thickness of 2 mm to 5 mm, which included the whole periderm and partial cortex structure (Figure 9b,c). After that, the skin tissues were cut into small pieces (Figure 9d), wrapped in aluminum foil paper, cooled down in liquid nitrogen, put into a 50 mL polypropylene conical tube, and stored in a −80 °C freezer for further usage. The processes are shown in Figure 9a–d. The total phenolic content was measured for all the skin samples, while the anthocyanin content was tested for the normal and silver patch skin samples.

### 3.3. Evaluation of Suberized Cell Layer

After the visual assessments of the tubers were completed, 4 medium-sized tubers were randomly picked out from each treatment plot and used for suberized cell layer analysis based on the method published by Dr. Gefu Wang-Pruski’s lab [20]. These tubers were washed, dried, and photographed on both sides using a digital camera (Sony DSC-F717) or a mobile phone (iPhone 13 Pro). Tuber skins were hand sliced into about 3 × 4 mm skin samples. The skin slices were stained with TBO solution (0.05% (*w*/*v*) Toluidine Blue O dissolved in 0.1 M sodium acetate (pH 4.5)), and then the samples were placed in complete darkness for 5–10 min. After that, the slices were washed with ddH_2_O and post-stained by neutral red (NR) solution (0.1% (*w*/*v*) Neutral Red dissolved in 0.1 M potassium phosphate (pH 6.5)) for 1–5 min. The stained slices were washed with ddH_2_O, de-stained by lactic acid (85% lactic acid and ddH_2_O water at 1:1 (*v*/*v*) ratio) and washed with ddH_2_O again. The prepared samples were observed under a microscope (Leica DMi8) and fluorescence light source (Leica EL6000) under a 10× objective lens. A Leica microscope and Leica DMC6200 camera were used to observe the samples and take images (Figure 10a). Based on the images, the number of suberized cell layers was counted (Figure 10b).

### 3.4. Determination of Total Phenolics

The total phenolics were measured using the Folin–Ciocalteu (FC) method [21] with garlic acid as a standard. The absorbance against the prepared sample reagents was measured using a UV-VIS spectrophotometer (Ultrospec 3000, Biochrom, Unit 7, Enterprise Zone, 3970 Cambridge Research Park, Beach Drive, Waterbeach, Cambridge, UK). Approximately 0.5 g of each the tuber skin samples was weighed and recorded. The absorbance was measured against a prepared reagent blank (0 mg/L gallic acid) at 760 nm. All samples were analyzed in duplicate. The total phenolic content was expressed as ‘mg gallic acid equivalents/100 g fresh weight’ (mg GAE/100 g fw). Based on the skin blemish occurrence, at most, 32 samples were measured for each skin type from 256 tubers each year. Among all four skin types, at most, 128 samples were measured from both the rainfed and irrigated treatments each year.

### 3.5. Determination of Total Anthocyanin

The anthocyanin content was analyzed using both visual observation and biochemical analysis to show if the reduced redness in the tuber skin colour was related to a loss of anthocyanin. Brightfield microscopy observation was performed for the colour comparisons. For the biochemical analysis, the anthocyanin contents in the normal and silver patch skins were evaluated in all treatments. The extraction and quantification of anthocyanin were carried out by the pH differential method [22], with a few modifications as indicated below.

The amount of sample per extraction was 100 mg in this experiment. The density of the skin tissue was set as 1 g/mL. The total dilution factor (DF) was determined to be 100 as shown in the equation:Total/Final dilution factor (DF) = DF_1_ × DF_2_ = 10 × 10 = 100(2)

The previously identified dilution factor (DF_1_) was set to be 10:DF_1_ = V_(tissue + solvent)_/V_(tissue)_ = 10(3)

The absorbance of the sample was read at 520 nm and 700 nm 3 times after zeroization.

The absorbance (A) and the total monomeric anthocyanin of each sample were calculated by using the equations:A = (Abs_520_ − Abs_700_)_pH 1.0_ − (Abs_520_ − Abs_700_)_pH 4.5_(4)
Total anthocyanin (mg C3GE/L) = (A × MW × DF × 1000)/(ε × L)(5)
where MW is the molecular weight of the predominant anthocyanin. In this experiment, cyanidin-3-O-glucoside (C3G) was used to express the total anthocyanin, since it is the most abundant anthocyanin in nature [23]. The MW of C3G is 449.2 g/mol. The molar extinction coefficient (ε) is 26,900. L is the path length (in cm), which is 1 cm. The conversion factor from g to mg is 1000. C3GE is the cyanidin-3-O-glucoside equivalent. The total anthocyanin (mg C3GE/L) was divided by 1000 to obtain a final unit of mg C3GE/100 mg fresh weight.

At most, 32 samples were measured for both the normal and silver patch skin types from 256 tubers each year.

### 3.6. Statistical Analysis

For total yield analysis, a two-sample *t*-test and Fisher’s least significant difference (LSD) pairwise comparisons were used. Mood’s Median test and Fisher’s LSD pairwise comparisons were used for skin blemish analysis. The statistically significant level was set as *p* = 0.05.

For suberin analysis, the number of suberized cell layers of all normal skin samples was compared between the rainfed and irrigated plots using a two-sample *t*-test. The number of suberized cell layers was compared among the 4 skin types using one-way ANOVA and Tukey’s pairwise comparison. A normality test was performed before the ANOVA and two-sample *t*-test. The statistically significant level was set as *p* = 0.05.

One-way ANOVA and Tukey’s pairwise comparison were used for the comparison of the total phenolic contents among 4 skin types, including normal, silver patch, surface cracking, and russeting. A two-sample *t*-test was performed to compare the total phenolic content from all the skin samples between rainfed and irrigated plots. A normality test was performed before the ANOVA and two-sample *t*-test. The statistically significant level was set as *p* = 0.05.

A two-sample *t*-test was performed to compare the total anthocyanin content in the normal and silver patch skin samples for both the 2020 and 2021 samples. A two-sample *t*-test was performed to compare all the normal and silver patch skin samples extracted in these 2 years. A normality test was performed before the two-sample *t*-test. When the data did not fit the normality, the Mann–Whitney test was used. The statistically significant level was set as *p* = 0.05.

## 4. Discussion

Water regime is an important factor that can affect tuber yield and quality. Irrigation did increase the total tuber yield and decreased the occurrence of surface cracking skin defects. The data from the 3-year field trials show that irrigated plants had significantly higher total yields (Figure 2). The yields of medium (2.25–3.0”) and large (3–3.5”) tubers were increased with irrigation. This result agrees with those of many previous studies about the importance of water availability during the growing season, especially its significant effect on tuber yield [24,25,26].

Surface cracking defects were found to be induced by water deficit in the field trial (Figure 3). Irrigation significantly reduced the occurrence (%) of surface cracking defects, however, russeting defects were increased to some degree (Figure 3). The higher russeting defects in the irrigated plots may have been caused by the expansion of the tuber skin in the skin developmental process [27], which is similar to the out-of-step cell division speed due to a fluctuating moisture supply [15].

The rainfed normal tubers tended to form more suberized cell layers compared to the irrigated normal tubers. Based on the three years of suberin analysis data, we found that the rainfed normal samples had more suberized cell layers than those of the irrigated normal samples. This result demonstrates that the tubers grown without irrigation tended to form more suberized cell layers, which can result in a thicker phellem. Following suberization, phellem cells die and create an outer defensive layer, which possesses a waxy component that protects against cell desiccation, and a protective suberin biopolymer, which provides a barrier to pathogens and other intrusions [7,28,29]. Suberin serves as a protective barrier in the periderm tissue layers, controls water and ion transport, restricts infection, and maintains integrity [7]. It has been reported that suberization in potato tuber periderm is associated with protection against biotic and abiotic stresses [7,27,28,29]. It has been observed that in response to heat stress, there was increased production and accumulation of periderm cell layers to protect the tubers, and many transcriptional factors of periderm responded to heat stress [7,30]. An increased number of suberized phellem cell layers also provided resistance against tuber greening [7,31]. Our results demonstrate that suberization can also be responsive to water deficit stress, during which the rainfed tubers tended to form more suberized cell layers to protect the tubers from water loss and a drought environment.

Many studies have been performed to understand the molecular mechanism of potato periderm [5,7,32,33]. A gene called ‘*CYP86A33*’ was proven to have a strong function in the formation of ω-functionalized monomers in aliphatic suberin, which are necessary for the suberin typical lamellar organization and the periderm resistance to water loss [34,35,36]. Another potato gene encoding a fatty ω-hydroxyacid/fatty alcohol hydroxycinnamoyl transferase (*FHT*) was reported to have significant effects on the anatomy, sealing properties, and maturation of the periderm [37]. When *FHT* was down-regulated, the tuber skin became thicker and russeted, water loss was greatly increased, and maturation was prevented [37]. It is suggested that future studies analyze the suberization-related gene expression under different water regimes, which can improve our understanding of the influence of water stress.

In addition to this, our results also show that the surface cracking and silver patch skins had more suberized cell layers than those of the normal and russeting skins. This result demonstrates that the normal skin had fewer suberized cell layers than skins with defects. However, this is opposite to a previous study that showed russeting had increased suberization and a thicker layer of phellem [27]. This difference could have been caused by the different observation methods and different potato varieties. Dark Red Norland is a smooth-skinned cultivar with a relatively thin phellem layer. The cells tend to be cracked when the suberization activity is increased. As the tuber skin expands during development, the thick part of the skin cracks away from the original thin skin and sloughs off, resulting in netted, rough skins. Surface cracking has a similar process of formation, which could also explain why the phellem of surface cracking seemed to be cracked and had more suberized cell layers (Figure 4 and Table 2). It is suggested that irrigation is applied throughout the tuber growing stage to reduce the soil temperature and create a good condition for Dark Red Norland tuber skin formation. Soil temperature is another important factor that can influence tuber growth, which is related to heat stress. A study showed that high temperature had negative effects on tuber yield and skin formation [38].

Significantly higher amounts of total phenolics were found in the irrigated treatments. Many studies have proven that environmental factors can profoundly influence the phenolic content in plants [8,21,39,40,41,42,43]. However, the results are often conflicting. Drought is likely to make plants accumulate phenolic compounds. The biosynthesis and accumulation of phenolic compounds during drought stress are regulated by enzymes of the phenylpropanoid pathway [42]. Studies on leafy lettuce, grapes, leaves of maize, and leaves of *Amaranthus tricolor* have observed a high accumulation of phenolic compounds in samples under drought stress [39,41,44,45]. In addition, cherry tomato, which also belongs to *Solanum* genus, was indicated to have decreased polyphenol content under irrigation. In contrast, contradictory results were obtained by Sánchez-Rodríguez et al. [46]. Studies on broccoli, sweet potato, and cauliflower have demonstrated that irrigation could have a positive effect on the phenolic content [43,47,48,49,50]. According to a review on the influence of water stress on the production of phenolic compounds in plants of medicinal interest [51], the widely accepted idea that there is a widespread increase in phenolic compounds in response to water stress is most often incorrect [51]. The total phenolic system is complicated and can be different for each plant species [51]. Our results show that irrigation had a positive effect on the phenolic content accumulation in Dark Red Norland tuber skins. This suggests that irrigation has an important role in regulating total phenolic biosynthesis in potato skins.

Significantly lower anthocyanin contents were found in silver patch skins (Figure 7). This finding is shown in Figure 6, in which less reddish pigmentation in the silver patch skins can be seen; instead, more brown-coloured compounds were found in the skin cells. It has been reported that phenolic extracts can strongly stimulate the oxidation of anthocyanin due to an anthocyanin-PPO (polyphenol oxidases)–phenol reaction that produces a brown by-product [52,53]. This explains why there were dark brown-coloured skin surfaces in the silver patch skin and not the pinkish-red skin colour; there was a higher concentration of total phenolics and less anthocyanin in the silver patch skins.

## 5. Conclusions

Irrigation plays an important role in Dark Red Norland potato production. It significantly improves the total yield of Dark Red Norland tubers, while reducing the occurrence of surface cracking skin blemishes and producing more good tubers. Our results also show that tubers grown without irrigation tended to form more suberized cell layers to protect the tubers from drought stress. Irrigation can increase the level of total phenolics in Dark Red Norland tuber skins. Different skin blemishes have different levels of suberization, total phenolics, and anthocyanins, in which surface cracking and silver patch form more suberized cell layers. Silver patch skins have higher contents of total phenolics but fewer anthocyanins. This study provides a better understanding of potato production and skin blemishes in Dark Red Norland tubers. Future studies can be conducted regarding drought stress effects on suberization-related genes in potato skins.

## Figures and Tables

**Figure 1 plants-11-03523-f001:**

Tuber skin blemishes found on Dark Red Norland, as shown in the marked areas. (**a**) Surface cracking (SC); (**b**) silver patch (SP); (**c**) russeting (R).

**Figure 2 plants-11-03523-f002:**
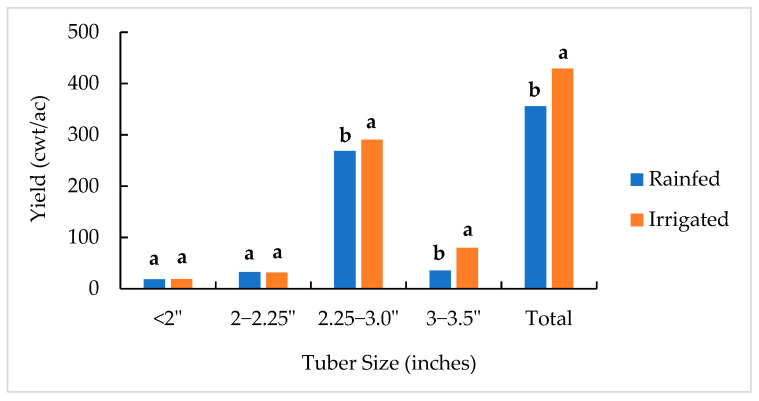
Tuber yields (cwt/ac) under rainfed and irrigated plots in different tuber sizes (in inches) of the three-year trial. Means not sharing a common letter in two adjacent columns are significantly different at *p* < 0.05 according to the Fisher LSD method.

**Figure 3 plants-11-03523-f003:**
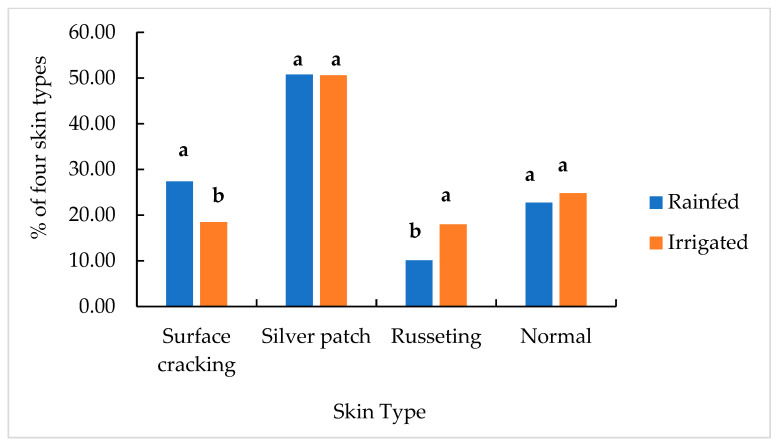
Percentages of the four skin types including ‘surface cracking’, ‘silver patch’, ‘russeting’, and ‘normal’ in all three years from rainfed and irrigated trials. Means not sharing a common letter in two adjacent columns are significantly different at *p* < 0.05 according to the Fisher LSD method.

**Figure 4 plants-11-03523-f004:**
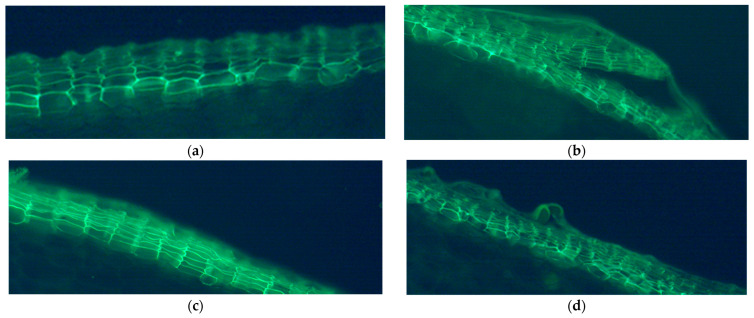
Representative samples of suberized cell layers in the (**a**) normal (N) skin, as well as the three defected skin types, including (**b**) surface cracking (SC), (**c**) silver patch (SP), and (**d**) russeting (R) of Dark Red Norland tubers under fluorescence. Photos were taken by Manlin Jiang using Leica LAS X Imaging and Analysis Software.

**Figure 5 plants-11-03523-f005:**
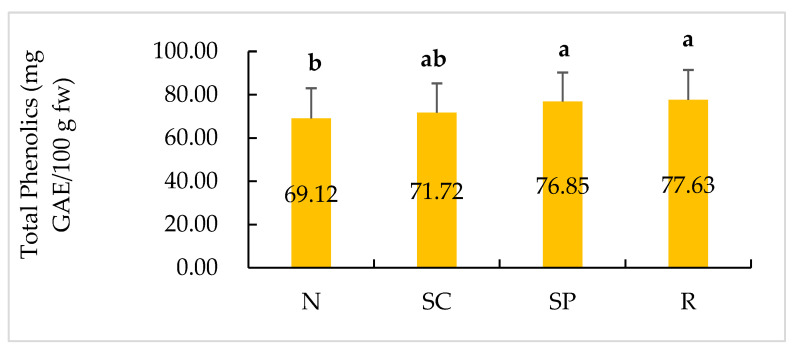
Comparisons of total phenolics (mg GAE/100 g fw) among four skin types including, normal (N), surface cracking (SC), silver patch (SP), and russeting (R) in two-year samples. Means that do not share a common letter are significantly different.

**Figure 6 plants-11-03523-f006:**
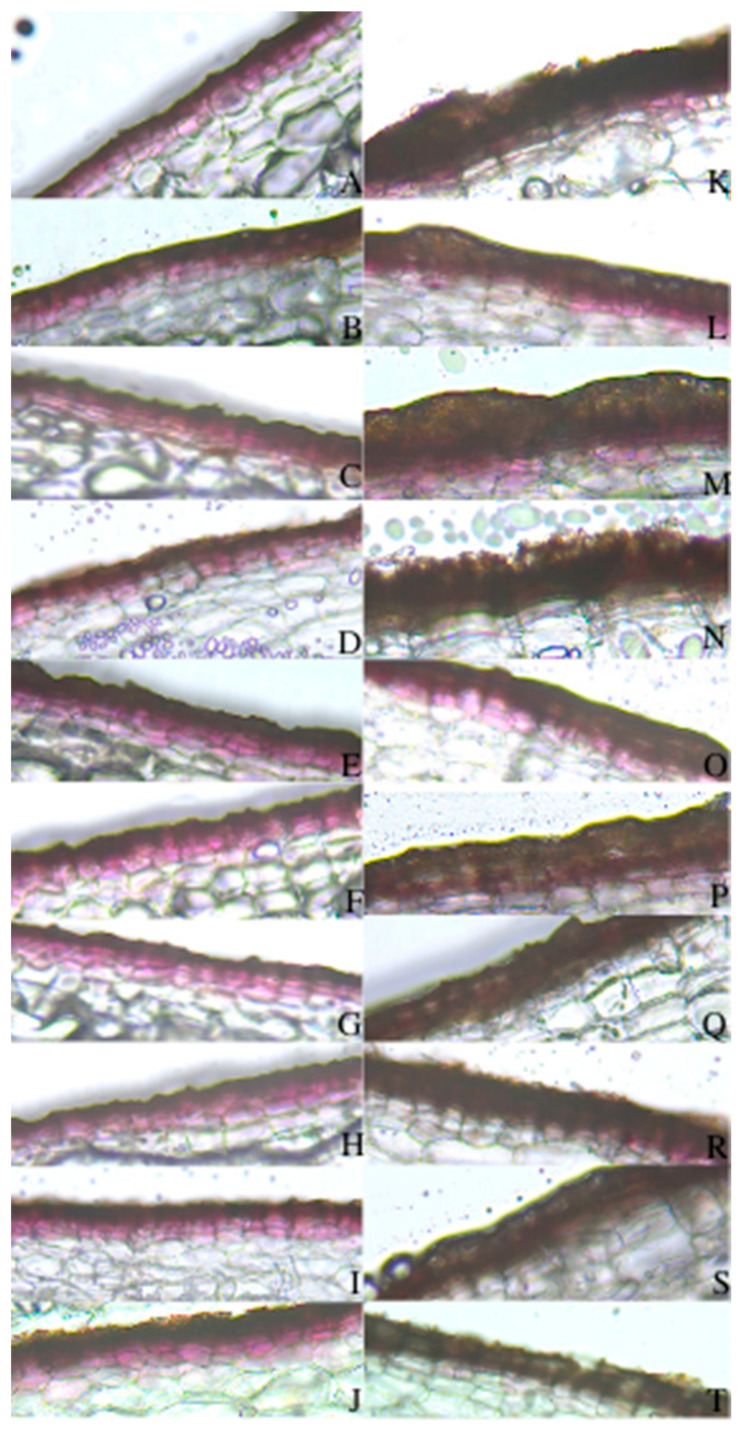
Photos comparison between normal skin (**A**–**J** on the left column) and silver patch skin (**K**–**T** on the right column) of Dark Red Norland tubers under bright field microscopy observation.

**Figure 7 plants-11-03523-f007:**
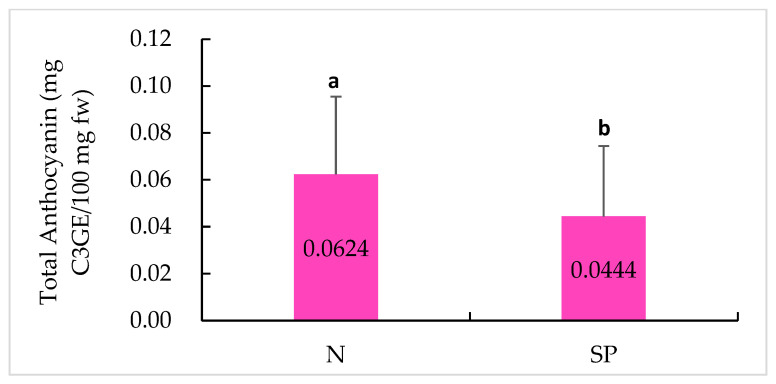
Comparisons of total anthocyanin contents between normal (N) and silver patch (SP) skins in all samples from 2020 and 2021. Means that do not share a common letter are significantly different at *p* = 0.05.

**Figure 8 plants-11-03523-f008:**
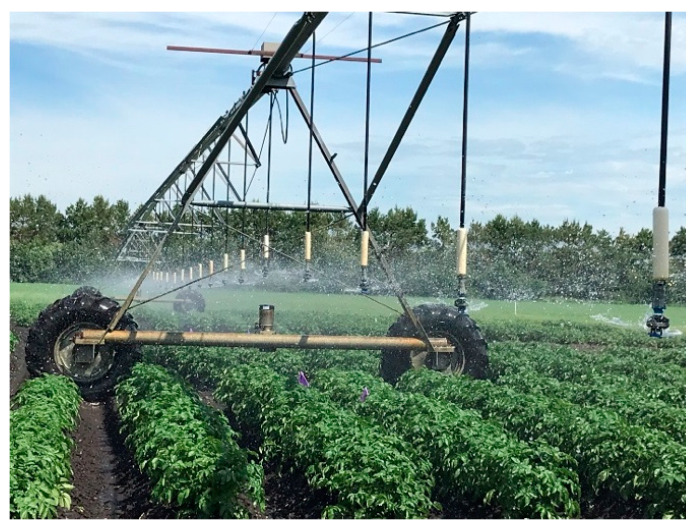
Irrigation system used in the field trials in Manitoba, Canada (photo was taken by Dr. Tracy Shinners-Carnelley).

**Figure 9 plants-11-03523-f009:**
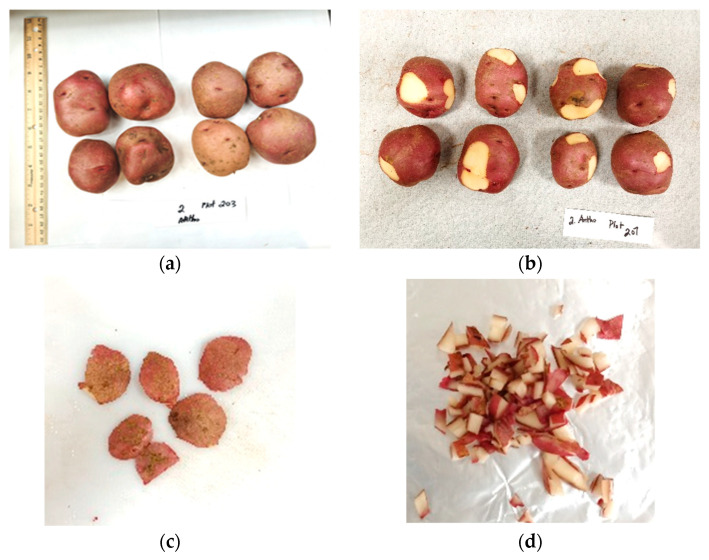
Photographs of sampling of selected tubers. (**a**) Eight tubers, including four tubers with relatively healthy skin type (left four showing red skin) and four tubers with blemishes of surface cracking, silver patch, and/or russeting (right four showing less reddish colour). (**b**,**c**) Skin samples were cut off from the eight tubers. (**d**) Skin samples were cut into small pieces.

**Figure 10 plants-11-03523-f010:**
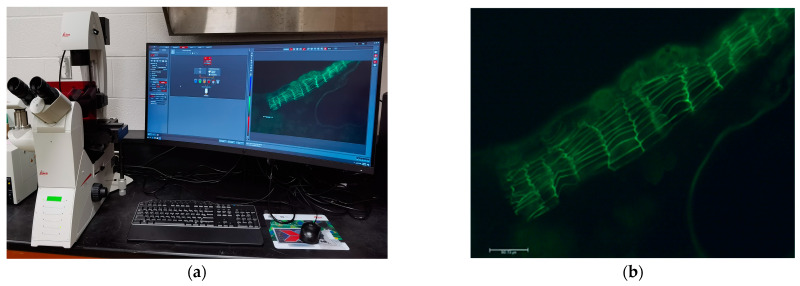
Analysis of the suberized cell layer. (**a**) The Lecia microscope (Leica DMi8) and Leica external light source for fluorescence excitation (Leica EL6000) and camera system (Leica DMC6200); (**b**) Image of the suberized cell layers under the fluorescence (captured by Leica LAS X Imaging and Analysis Software), showing the number of layers of suberized cells. (Leica system can be found in Leica Microsystems Inc., 71 Four Valley Drive, Concord, ON, Canada.)

**Table 1 plants-11-03523-t001:** Two-sample *t*-test comparisons of suberized cell layer in normal skin samples between two water regimes in each year.

Water Regime	2019	2020	2021
Rainfed	7.327 ^a^	6.730 ^a^	7.110 ^a^
Irrigated	6.675 ^b^	6.370 ^b^	6.735 ^b^

Means that do not share a letter within a column are significantly different, *p* = 0.05.

**Table 2 plants-11-03523-t002:** Pairwise comparisons of suberized cell layer using Tukey’s method among four skin types, including normal skin (N), surface cracking (SC), silver patch (SP), and russeting (R) in samples of 3 years.

Skin Type	Treatment							
	Rainfed				Irrigated			
	No Stress	Tuber Initiation	Tuber Bulking	Tuber Skinset	No Stress	Tuber Initiation	Tuber Bulking	Tuber Skinset
N	7.000 ^c^	6.900 ^b^	6.959 ^b^	7.358 ^b^	6.752 ^c^	6.385 ^c^	6.600 ^c^	6.611 ^b^
SC	7.856 ^b^	7.970 ^a^	7.805 ^a^	8.330 ^a^	8.329 ^a^	8.170 ^a^	8.333 ^a^	8.031 ^a^
SP	8.495 ^a^	8.000 ^a^	7.740 ^a^	8.086 ^a^	7.835 ^b^	7.950 ^a^	7.710 ^b^	8.119 ^a^
R	6.464 ^c^	6.282 ^c^	6.860 ^b^	6.700 ^c^	7.000 ^c^	7.044 ^b^	6.810 ^c^	6.214 ^b^

Means that do not share a letter within a column are significantly different, *p* = 0.05.

**Table 3 plants-11-03523-t003:** Irrigation dates and applied amount (inches) in each year from 2019 to 2021.

Year	Date of Irrigation	Amount Applied (Inches)
2019	3 June	0.50
	5 June	0.50
	12 June	0.50
	27 June	0.50
	3 July	0.50
	18 July	0.50
	24 July	0.50
	29 July	0.50
	5 August	0.75
	7 August	0.50
	13 August	0.75
2020	4 June	0.75
	17 June	0.75
	24 June	0.50
	29 June	0.50
	21 July	0.75
	29 July	1.20
2021	24 June	1.00
	30 June	0.75
	6 July	0.50
	9 July	0.50

## Data Availability

All the relevant data of the study are provided in the manuscript.

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
