# Peer review of "Irrigation Effect on Yield, Skin Blemishes, Phellem Formation, and Total Phenolics of Red Potatoes"

_plants, 2022, doi:10.3390/plants11243523_

Round 1

Reviewer 1 Report

The manuscript “Irrigation effects on yield, skin blemishes, phellem formation, and total phenolic of red potatoes'

A three-year field trial was conducted by Peak of the Market in Manitoba, Canada and potato plants were treated under both irrigation and rainfed conditions. Results showed that irrigation increased total yield and reduced  the severity of surface cracking. 

The topic is well determined and red potatoes must be studied in overall world because the plant has some unique  properties. 

The manuscript is prepared professionally. It includes a well-crafted abstract and an exhaustive introduction that justifies the research undertaken. The introduction points to the deficiencies in the literature on the subject. The aim is clearly defined. Modern analytical methods were used in the research. The discussion of the results is well prepared. The conclusions are well-defined. The illustrative material is appropriate.

In my opinion, the manuscript after corrections, will be suitable for publication in a journal.

Detailed comments:

Abstract: Should include more numeric data from obtained results.

Introduction - The introduction is enough in my opinion but could be improve by using new sentences and related references

Line 29 Please correct sentence as 

Potato (Solanum tuberosum L.) is the fourth most important crop in terms of volume and consumption after maize, wheat and rice in the world. It also one of the most studied crops in literature  [1-3]. 

Please add more references. I suggest below ones. 

Asim, A.; Gokce, Z.N.O.; Baksh, A.; Cayli, I.T.; Aksoy, E.; Caliskan, S.; Caliskan, M.E.; Demirel, U . Individual and combined effect of drought and heat stresses in contrasting potato cultivars over expressing miR172b-3p. Turk. J. Agric. For. 2021, 45 (5): 651-668.

Hussain, M.J.; Aksoy, E.; Gokce, N.Z.O.; Joyia, F.A.; Khan, M.S.; Bakhsh, A (2021). Rapid and efficient in vitro regeneration of transplastomic potato (Solanum tuberosun L.) plants after particle bombardment. Turk. J. Agric. For. 45 (3):313-323.

Please add more references. I suggest below ones. 

Why irrigation schedule in different years?

Why choose C3GE? Any special reason?

The study showed that rainfed normal tubers tend to form more suberized cell layers compared with irrigated normal tubers. Based on three-year suberin analysis data, they found that rainfed normal samples had more suberized cell layers than that of irrigated normal samples. This result demonstrates that tubers grown without irrigation tend to form more suberized cell layers, which can result in a thicker phellem. 

Please explain the physiological or molecular mechanism of this?

Please explain mechanism why irrigated crops had higher total phenolic content? Please search literature on this topic. 

Figure 9 is not well informative. 

Reviewer 2 Report

Dear Authors,

I am glad that I had the opportunity to review the manuscript entitled: "Irrigation Effect on Yield, Skin Blemishes, Phellem Formation, and Total Phenolics of Red Potatoes" by Manlin Jiang, Tracy Shinners-Carnelley, Darin Gibson, Debbie Jones, Jyoti Joshi and Gefu Wang- Prussian".

The aim of the research presented in this manuscript was to investigate the relationships between water regime and the yield of 'Dark Red Norland' potato, and tuber skin blemishes.

Undoubtedly, the subject of research and the results obtained by the Authors are interesting and fully deserve to be published. However, the manuscript needs some refinements. Below are some comments to which the Authors should respond:

1) There is no hypothesis in the Abstract, please complete it.

2) Keywords should be in alphabetical order and should not repeat words that are already in the title of the manuscript.

3) L 98 - "The field trial to produce Dark Red Norland tubers for this study was conducted..." please add here the words "potato cultivar" and the Latin name of the species.

4) L 116 - please prepare all formulas in accordance with the guidelines of the template.

5) L 219, 231, 285, 289, 322 - please use "p" instead of "α".

6) L 245 - please use "p" instead of "P".

7) Table 3 - please explain the abbreviations N, SC, SP and R. Please note that all tables and figures should be understood without having to look in the manuscript text for an explanation of the abbreviations used.

8) Figure 8 - L 299 - please add: N, SC, SP and R.

9) The discussion is poorly developed, moreover, I also propose to extend the list of cited publications.

10) I think it is necessary to add a Conclusions chapter.

11) Please adapt the entire manuscript more carefully to the requirements of the template in force in the journal Plants, especially the References chapter.

In conclusion, I believe that Plants should consider publishing this manuscript.

Round 2

Reviewer 1 Report

The authors responded to all my comments.